# Sodium Intake as a Cardiovascular Risk Factor: A Narrative Review

**DOI:** 10.3390/nu13093177

**Published:** 2021-09-12

**Authors:** David A. Jaques, Gregoire Wuerzner, Belen Ponte

**Affiliations:** 1Division of Nephrology and Hypertension, Department of Medicine, Geneva University Hospitals, 1205 Geneva, Switzerland; belen.ponte@hcuge.ch; 2Division of Nephrology and Hypertension, Department of Medicine, Lausanne University Hospitals, 1011 Lausanne, Switzerland; gregoire.wuerzner@chuv.ch

**Keywords:** salt, sodium, blood pressure, hypertension, cardiovascular

## Abstract

While sodium is essential for human homeostasis, current salt consumption far exceeds physiological needs. Strong evidence suggests a direct causal relationship between sodium intake and blood pressure (BP) and a modest reduction in salt consumption is associated with a meaningful reduction in BP in hypertensive as well as normotensive individuals. Moreover, while long-term randomized controlled trials are still lacking, it is reasonable to assume a direct relationship between sodium intake and cardiovascular outcomes. However, a consensus has yet to be reached on the effectiveness, safety and feasibility of sodium intake reduction on an individual level. Beyond indirect BP-mediated effects, detrimental consequences of high sodium intake are manifold and pathways involving vascular damage, oxidative stress, hormonal alterations, the immune system and the gut microbiome have been described. Globally, while individual response to salt intake is variable, sodium should be perceived as a cardiovascular risk factor when consumed in excess. Reduction of sodium intake on a population level thus presents a potential strategy to reduce the burden of cardiovascular disease worldwide. In this review, we provide an update on the consequences of salt intake on human health, focusing on BP and cardiovascular outcomes as well as underlying pathophysiological hypotheses.

## 1. Introduction

Sodium (Na+), contained in dietary salt, is essential for human homeostasis. For millions of years, our ancestors ate less than 0.25 g of salt per day, while the current average daily consumption approaches 10 g in most countries [1,2]. Such an increase over a comparatively modest time span imposes a significant physiological challenge in evolutionary terms. Excessive sodium intake is thought to adversely affect our health through effects on blood pressure (BP) and cardiovascular damages. Consequently, three million deaths were attributed to high salt intake in 2017 [3]. Given that the majority of cardiovascular burden affects individuals with high-normal BP or mild hypertension, dietary and lifestyle programs including salt reduction constitute attractive and simple public health measures [4]. Despite general agreement that excessive sodium consumption is globally harmful, controversies still exist on the net benefit of sodium intake reduction on a population level and the levels that should be targeted [5]. Moreover, far from a simplistic cause–effect relationship, pathophysiological mechanisms linking sodium intake and cardiovascular outcomes are diverse and intricate.

In this paper, we review the available evidence on the association between sodium intake, BP and cardiovascular diseases. We more specifically discuss major pathophysiological hypotheses underlying this relationship.

## 2. Sodium Intake, Blood Pressure and Cardiovascular Outcomes

### 2.1. Blood Pressure

Convincing evidence suggest a direct and positive association between sodium intake and BP regulation. At the population level, the *INTERSALT* (International Study of Sodium, Potassium and Blood Pressure) study was the first international study to look at this association [6]. This cross-sectional analysis described the relationship between sodium intake based on 24-h urine collection and BP in over 10,000 participants aged 20 to 50 from 39 countries. The authors reported a significant association between sodium excretion and BP at the individual level. Furthermore, sodium intake was also associated with age-related hypertension, suggesting that sodium could also have a long-term impact in addition to its immediate effect on BP regulation. More recently, the *PURE* (Prospective Urban Rural Epidemiology) study, a large international report including more than 100,000 adult participants from 18 countries, was published [7]. A positive curvilinear relationship between sodium intake and BP was described. In line with *INTERSALT*, this relationship was stronger in older individuals and those consuming low potassium diets. The *UK Biobank study* is the largest report to date with urinary electrolyte data and BP measurements in more than 450,000 adult subjects [8]. Authors found a positive linear association between urinary sodium excretion and BP. On top of those cross-sectional data, longitudinal studies are also available. In the *EPOGH* (European Project on Genes in Hypertension) study, investigators followed a group of 1499 participants without cardiovascular disease over 6.1 years [9]. Compared to baseline values, increased sodium intake was associated with an increase in BP after adjusting for potential confounders. In such observational studies, a systolic blood pressure increase of 2 to 3 mmHg for each 1 g/day increment in estimated sodium excretion was generally reported [7,8].

In addition to epidemiological studies, numerous randomized controlled trials have confirmed the impact of dietary sodium on BP values and control of hypertension (Table 1). The *DASH-sodium* (Dietary Approaches to Stop Hypertension) study included around 400 pre-hypertensive individuals [10]. Authors evaluated the impact on BP control of three different diets with various sodium intake (1.5 g/day, 2.4 g/day and 3.3 g/day) consumed for 30 days. On average, systolic BP values decreased by 2.1 mmHg when comparing 3.3 g/day to 2.4 g/day diets. An additional 4.6 mmHg decrease was reported when comparing 2.4 g/day to 1.5 g/day diets, highlighting a dose-dependent relationship between sodium consumption and BP regulation. In the *TOHP-II* (Trial of Hypertension Prevention) study, investigators evaluated the impact of long-term sodium intake reduction and weight loss on blood pressure in 2382 individuals not taking antihypertensive medications using a factorial design [11]. Despite a sodium intake target below 1.8 g/day, mean sodium intake was 3.1 g/day and 3.2 g/day at 18 and 36 months, respectively. In comparison, mean sodium intake was 4.0 g/day at 36 months in the control group. Consequently, the improvement in BP values and hypertension prevalence achieved in the intervention group decreased over time. As such, the primary efficacy endpoint, defined as the mean decrease in diastolic BP, was not significant at 36 months. The *TONE* (Trial of Nonpharmacological Intervention in the Elderly) trial also tested the impact of sodium reduction and weight loss on BP control and cardiovascular events in 975 treated hypertensive adults aged between 60 and 80 [12]. As in the *TOHP-II* study, the sodium intake target in the intervention group was 1.8 g/day. At 3-months follow-up, a 3.4 mmHg reduction in systolic BP was achieved in the intervention group. Thirty months later, a higher proportion of patients were withdrawn from antihypertensive medication in the intervention group compared to the control group.

Finally, meta-analyses of interventional trials globally confirmed an antihypertensive effect of sodium reduction. A first meta-analysis, which included 36 randomized controlled trials and 6736 adult individuals showed that reduced sodium intake was associated with a decrease in BP of 3.39/1.54 mmHg without an adverse effect on renal function, or metabolic or endocrine profile [13]. Of note, in sensitivity analyses, taking into account the study’s duration, the authors reported a decrease in this beneficial effect with longer follow-up. A second meta-analysis included randomized controlled trials investigating a modest reduction in sodium intake with a minimum follow-up of four weeks [14]. A total of 34 trials and 3230 participants were identified. Investigators showed that a sodium intake reduction of 4.4 g/day decreased BP of 5.4/2.8 mmHg in hypertensive individuals and 2.4/1.0 mmHg in normotensive individuals.

Although the beneficial effect on BP is clear, it has been previously suggested that sodium intake reduction could potentially lead to harmful consequences. A meta-analysis of 167 trials randomizing patients to low versus high sodium diets concluded that sodium reduction resulted in an increase in renin, aldosterone, catecholamine and cholesterol levels [15]. However, this report included studies involving large reductions in sodium intake over a very short period of time. Such an abrupt decrease in sodium consumption is expected to enhance several compensatory mechanisms, in contrast to what has been described with a modest reduction over longer periods [16]. Second, a dose-response relationship between sodium intake and BP has consistently been shown across observational as well as interventional trials, suggesting that benefits of sodium reduction could extend to very low values [6,10,17,18]. Third, studies have generally shown that sodium reduction allowed a significantly greater BP fall in older, hypertensive and Afro-American individuals [6,12]. Such findings can be linked to varying sensitivities of the renin–angiotensin–aldosterone system (RAAS) regulation in different populations [19,20]. Finally, sodium reduction has synergistic effects with pharmacological and conservative measures on BP control. The DASH-sodium trial showed that a combination of low sodium and a healthy diet has a greater effect on BP reduction than individual measures [10]. The *TONE* and *TOHP-II* trials reported similar additive effects of sodium reduction with weight reduction [11,12]. Taking into account such compensatory mechanism, a randomized controlled trial showed that sodium reduction allowed for a further reduction of BP in hypertensive individuals treated with captopril as compared to normal sodium intake [21].

Sodium intake has varying effects on different subjects. In its simplest definition “salt sensitivity” is a physiological trait by which BP exhibits changes parallel to sodium intake [22]. Conversely, in “salt resistant” individuals, BP does not vary according to salt loading. In humans, salt sensitivity is a continuous, normally distributed, quantitative characteristic and any distinction between salt sensitive and salt resistant individuals is, thus, somewhat arbitrary [23]. Although the definition and identification of salt sensitivity lacks uniformity, it is suggested that up to 50% of hypertensive and 25% of normotensive individuals are salt sensitive [24]. Overall, the elderly, women and those with Afro-American ethnicity, as well as patients suffering from chronic kidney disease, diabetes and primary aldosteronism, are more prone to salt sensitivity [25,26,27]. Given the pathophysiological complexity of hypertension, identification of genetic factors associated with salt sensitivity is an intensive field of research [28]. Gene–environment interactions have been characterized where polymorphisms in RAAS genes modified the effect of sodium intake on BP in Japanese workers [29]. Such considerations could partially explain heterogeneous findings in clinical studies as well as facilitate personalized approaches to sodium intake recommendations.

Globally, the causal relationship between sodium intake and BP control is, thus, well established, and modest reduction in salt consumption is associated with a meaningful reduction in BP on a population level. However, as the full effect of sodium intake reduction on BP control is not reached until several weeks, results of interventional trials could have underestimated the magnitude of this effect as most studies lasted a few weeks only [14,30]. On the other hand, the few existing long-term interventional trials showed that maintaining a lower salt intake on a prolonged time period is challenging from an individual perspective given the societal food environment [11,31].

### 2.2. Cardiovascular Outcomes

As sodium intake reduction lowers BP in normotensive and hypertensive individuals, it could also be expected to improve cardiovascular outcomes. However, evidence from large, long-term, randomized controlled trials on an individual level is currently lacking.

Over the years, numerous cohort studies have been published exploring the relationship between sodium intake and cardiovascular outcomes. Several meta-analyses have pooled those studies and come to the conclusion that salt consumption directly and negatively impacts cardiovascular prognosis [13,32,33]. Interestingly, several independent cohort studies have reported a U-shaped association where both low and high sodium consumption increased the risk of cardiovascular events and mortality when compared with moderate intake [34,35,36,37]. Despite this, major potential confounding factors impede definite conclusions and a causative association between low sodium intake and increased cardiovascular risk for several reasons. First, reverse causality likely introduces a significant bias as those studies included subjects at high cardiovascular risk, probably already advised to lower their salt consumption. The increased morbidity associated with decreased sodium intake might therefore represent the confounding effect of underlying cardiovascular diseases. Second, most of those studies relied on a single morning fasting urine sample to estimate 24 h sodium consumption. Kawasaki’s formula is usually used in such circumstances, including parameters such as age, gender, height and weight [38]. As these variables are themselves potentially associated with the adverse prognosis, this could further confound the association between sodium intake and the considered outcomes. As such, 24 h urinary collection should be considered the gold standard to reliably estimate sodium intake in such a setting. Globally, cohort studies that relied on multiple 24 h urinary collections universally reported a direct positive linear relationship between sodium intake and cardiovascular events [39,40,41,42]. Moreover, post hoc analysis of the data from the *TOHP trial* has shown that the association between mortality and sodium intake was J or U-shaped only when using formula-derived spot estimates but linear when measured on 24 h urine collections [43]. In agreement with these reports, an observational study recently showed that 24 h estimates of sodium consumption derived from formulae based on spot urine measurement overestimated sodium consumption at lower levels and underestimated sodium consumption at higher levels when compared to directly measured 24 h sodium urinary excretion [44]. Consequently, while a linear association existed between systolic BP and measured 24 h sodium excretion, formula-derived estimates resulted in a J-shaped association, thus altering the true relationship between dietary sodium and BP.

As patients with chronic kidney disease are at increased risk of cardiovascular events compared with the general population, the effect of sodium consumption in this specific sub-group is of pivotal importance. To answer this question, a prospective cohort study included more than 3700 participants with chronic kidney disease and evaluated the association between urinary sodium excretion and clinical cardiovascular events with a median follow-up of 6.8 years [41]. Higher sodium consumption was associated with increased risk of cardiovascular events in this population. Patients suffering from heart disease also constitute a sub-group of special relevance. Left ventricular hypertrophy is considered a major predictor of cardiovascular prognosis and, since left ventricular hypertrophy is directly caused by hypertension, it could be expected that salt consumption would indirectly induce ventricular remodelling through raised BP [45]. However, there exists increasing evidence suggesting that sodium intake may directly induce left ventricular hypertrophy, independent of BP [46,47]. A review of nine cross-sectional studies described a strong correlation between sodium intake as assessed by 24 h urinary collection and left ventricular mass, independent of relevant confounders as well as BP [48]. In a prospective cohort study including more than 10,000 participants, higher salt consumption was associated with increased risk of developing heart failure over a follow-up of almost 20 years [49]. This association was, however, significant in overweight individuals only. Conversely, a low salt diet is a mainstay of congestive heart failure management. As such, in a cohort of 443 patients with preserved ejection fraction, those who were recommended a sodium-restricted diet had lower risk of death and hospital readmission at 30 days [50].

Estimation of sodium consumption on a population level is more reliable than at an individual level [51]. Overall, evidence derived from public health interventions describe a robust relationship between sodium intake and cardiovascular outcomes. For example, Finland, and more recently the UK, have led salt reduction campaigns in an effort to improve national health and both were successful at significantly reducing BP and cardiovascular burden over the course of several years [52,53]. In Finland, salt consumption decreased from 14 g/day in 1972 to 9 g/day in 2002 [53]. During this time period, mean systolic and diastolic BP decreased by 10 mmHg and cardiovascular mortality by 75%. While other factors could have played a role, obesity and alcohol consumption have both increased during this time period suggesting a pivotal impact of sodium restriction. In the UK, salt consumption decreased from 9.5 g/day in 2003 to 8.1 g/day in 2011 [52] and a mean decrease in systolic BP of 2.7 mmHg was reported after adjustment for potential confounders, while stroke and ischemic heart disease mortality decreased by 36%.

Finally, in a cluster randomized controlled trial including five veteran’s retirement homes in Taiwan [54], sodium intake was reduced from 5.2 to 3.8 g/day in the intervention group in combination with increased potassium, resulting in a reduction in cardiovascular mortality. Follow-up studies have also been conducted including individuals who previously participated in randomized controlled trials of sodium intake reduction (Table 1), later aggregated in meta-analyses. Despite including seven studies pooling 6489 participants with or without hypertension, the first study found no significant association between reduced sodium intake and cardiovascular events [55]. Authors concluded that there is insufficient power to exclude a clinically important effect of reduced sodium consumption on cardiovascular morbidity and called for future, large, long-term, randomized controlled trials focusing on clinical outcomes. According to their own calculation, 2500 cardiovascular events would be needed to achieve sufficient power to detect a clinically meaningful effect. Such a trial would require thousands of participants adhering to specific sodium diets over several years. Imposing a high salt diet on a large scale for an extended period would raise serious ethical and methodological concerns, rendering such a study unlikely to ever be conducted. A second meta-analysis conducted by another group later reanalysed the same dataset in with a slightly different methodology [56]. Indeed, one study was excluded from the pooled analysis as it involved heart failure patients who were likely to be sodium depleted before randomization as they had already been treated with furosemide, spironolactone, captopril and fluid restriction [57]. Then, instead of considering two distinct groups of patients, normotensive and hypertensive individuals were analysed collectively, thereby increasing statistical power. By implementing those methodological modifications, authors showed that reduced sodium intake was associated with a significant decrease in cardiovascular events and a non-significant trend toward lower mortality.

Considering the available evidence, it is reasonable to assume a direct and continuous relationship between sodium intake, BP control and cardiovascular outcomes from a population standpoint. However, as definite evidence from long-term randomized controlled trials is still lacking, a consensus has yet to be reached on the effectiveness, safety and feasibility of sodium intake reduction on an individual level. Until then, it seems judicious to routinely evaluate sodium intake in patients at high cardio-renal risk and recommend adherence to low sodium diet as part of a multifaceted treatment approach in an effort to reduce morbidity and mortality in those patients [58,59].

## 3. Pathophysiological Considerations

### 3.1. Basic Principles

Multiple mechanisms are responsible for the association between sodium intake and BP. The key role of the kidney in this relationship has been clearly demonstrated in several transplant experiments [60,61]. Guyton formalized the basic principle of kidney BP regulation as the pressure natriuresis response [62]. In its simplest form, this concept states that when BP increases, the excess pressure causes the kidney to excrete more sodium and water. This, subsequently, decreases extracellular blood volume and, thus, preload as well as cardiac output, thereby restoring BP to lower levels. On the other hand, variations in sodium intake induce parallel changes in plasma sodium content, both in hypertensive and normotensive individuals [63,64]. A rise in plasma sodium increases osmolarity, thus inducing a fluid shift from the intracellular to the extracellular compartment. A small increase in plasma osmolarity also stimulates vasopressin secretion, thus resulting in water retention. Both these mechanisms restore plasma sodium to its original level but also increase extracellular fluid volume thereby increasing BP [65]. Importantly, evidence exists that a variation in plasma sodium can influence BP regulation independently of blood volume variation [66]. Such additive effects could be mediated by the direct influence of plasma sodium on the hypothalamus; the vasculature as well as the immune system [67,68,69].

### 3.2. Organ Damage and Cardiovascular Impact

Sodium intake indirectly affects target organs via its effect on BP. Hypertension is associated with endothelial dysfunction and platelet activation eventually resulting in microvascular as well as macrovascular disease and target organ damage [70]. However, evidence also points towards direct adverse consequences of sodium load on cardiovascular prognosis independent of BP (Figure 1). In this section, we review the main mechanisms involved in this pathophysiological process.

Excessive sodium has been shown to be involved in different pathways such as oxidative stress, inflammation and fibrosis, which are determinant in target organs being damaged. In patients with non-diabetic chronic kidney disease, sodium restriction increases the concentration of anti-inflammatory and anti-fibrotic peptides on top of RAAS blockade, providing pathophysiological insights into the synergistic benefit of sodium reduction and RAAS blockade [71]. Animal studies confirmed a direct pro-fibrotic effect of sodium on glomeruli mediated via an increased local expression of transforming growth factor (TGF) β [72,73]. Additionally, sodium intake directly influences nitric oxide (NO) generation and local oxidative stress in rats’ kidneys [74,75]. Thus, in middle-aged hypertensive adults, dietary sodium restriction largely reversed macro- and microvascular endothelial dysfunction by enhancing NO bioavailability and decreasing oxidative stress, thus supporting a direct vascular protective effect of sodium restriction beyond any influence on BP regulation [76]. 

Some evidence suggests that sodium is involved in cellular senescence as a high NaCl environment inhibits the activity of key components of the classical DNA damage response, such as Mre11 exonuclease in cell culture [77,78]. Exposure to NaCl was also associated, both in cell culture as well as in vivo, with increased senescence-associated β-galactosidase activity and p16^INK4^ expression as well as reduced levels of Hsp70, all of which are indicative of cellular senescence [79]. High salt intake is also associated with cell death in animal experiments as cytosolic accumulation of caspase-independent apoptotic factors, such as apoptosis-induced factor (AIF) and HtrA2/Omi, was described in response to sodium loading [80]. Finally, apoptosis signal-regulating kinase 1 (ASK1) has been involved in sodium-induced organ damage in animal models. ASK1 deficiency abolished sodium-induced cardiac and vascular pathological alterations in normotensive mice [81]. In a second experimental study, ASK1 deficiency also improved aldosterone/salt-induced cardiac inflammation and fibrosis. Furthermore, the enhancement of NADPH oxidase-induced cardiac oxidative stress caused by aldosterone and sodium was notably decreased by ASK1 deficiency [82]. A schematic representation of potential pathways involved in sodium-induced cellular injury is given in Figure 2.

In healthy adults, endothelial function, as measured by flow-mediated dilatation and arterial stiffness, was negatively impacted by high sodium intake [83,84], and reducing sodium intake had the opposite effect [85]. Importantly, the adverse influence of sodium on vascular function was unequivocally found to be independent of BP [86,87,88]. Sodium intake was also positively and independently associated with renal resistive index (RRI) in an adult general population [89]. As RRI could represent systemic vascular damage and correlates with adverse cardiovascular outcomes, this suggests that sodium consumption impacts renal hemodynamic as a reflection of a broader systemic alteration [90,91]. An effect of sodium intake on systemic vascular properties independent of BP has been confirmed in other studies. High sodium consumption was associated with increased arterial stiffness as measured by pulse wave velocity after adjustment for BP in Chinese communities [92]. The same group showed that pulse wave velocity was decreased in Australian normotensive subjects adhering to a low sodium diet compared with age and BP matched controls [93]. More recently, a meta-analysis comprising 11 randomized controlled trials and 14 independent cohorts reported a positive association between sodium intake and pulse wave velocity beyond BP control. Globally, those findings are all the more important in that the severity of endothelial dysfunction relates to the global cardiovascular risk [94]. On a cellular level, dietary salt has direct effects on vascular endothelium NO production [95]. In bovine aortic endothelial cells, sodium exposure caused a significant decrease in NO synthase (NOS) activity in a dose-dependent manner, thereby explaining a salt-induced reduction in endothelial NO generation [96]. High salt intake led to increased superoxide production and decreased NO bioavailability in mice aortas [97]. Furthermore, a high salt diet impaired aortic ring endothelium-dependent relaxation via reduced NO levels and increased superoxide production in rat aorta [98]. In spontaneously hypertensive rats, excessive dietary salt decreased cyclic GMP production in the aorta, leading to the impairment of NO-mediated vascular relaxation despite increased NO production [99]. Such alterations of the NO/cyclic GMP system were restored by dietary salt restriction but not by antihypertensive therapy in a later study [100]. Activation of vasopressor mechanisms were also described, as a high salt diet increased the expression of cytochrome p450 4A enzymes in rat mesenteric arteries, thus upregulating the production of 20-hydroxyeicosatetraenoic acid (20-HETE) [101]. In turn, 20-HETE contributed to the vasoconstrictor response to norepinephrine in those arteries [101]. A schematic representation of potential pathways involved in sodium-induced vascular alterations is given in Figure 3.

The adverse effect of sodium on target organs could be partially mediated by hormonal interactions. In a case-control study including 21 patients with primary aldosteronism and 21 hypertensive control patients, 24 h sodium excretion was associated with left ventricular mass and thickness only in patients with primary aldosteronism [102]. Aldosterone excess may thus play a permissive role in sodium induced target organ damage. Another study prospectively investigated the influence of sodium intake on cardiac outcomes of patients both before and after treatment of primary aldosteronism [103]. Interestingly, sodium intake interacted with aldosterone in inducing cardiac changes over time, while left ventricular mass was associated with both sodium intake and aldosterone levels before treatment. The decrease in ventricular mass obtained after treatment was greater in patients whose sodium intake also decreased. Change of ventricular mass was also associated with sodium intake independently of BP and other potential confounders. In another study including 90 adults with essential hypertension, left ventricular mass was associated with plasma aldosterone level after, but not prior to, intravenous saline load, implying that a limited ability of sodium to supress aldosterone production could contribute to organ damage [104]. Finally, in a longitudinal study, 182 adults with essential hypertension and left ventricular hypertrophy were treated with RAAS blockade [105]. The observed decrease in left ventricular mass over time was correlated with change in BP, 24-h sodium urinary excretion and plasma aldosterone concentration. At the end of the follow-up, the combination of high sodium intake and high aldosterone levels was associated with increased left ventricular mass. In contrast, in patients with low sodium intake, no influence of aldosterone levels was detected. These results together suggest that persistence of organ damage despite adequate BP control may result from the combined effect of excessive sodium intake and breakthrough of aldosterone despite pharmacological blockage. Basic research studies also support the interplay between sodium and aldosterone in organ damage physiopathology as aldosterone-induced superoxide over-production and vascular smooth muscle hypertrophy in cell culture, a phenomenon synergistically augmented with sodium chloride [106].

Finally, sodium could impact cardiovascular prognosis via other less studied pathways. Evidence suggests that sodium has immunomodulatory properties as interleukin-17 producing T lymphocytes are highly pro-inflammatory cells whose differentiation was enhanced in vitro and in vivo by a modest sodium load [107]. Extensive renal T cell infiltration has otherwise been described in numerous in vivo experiments under a high salt diet [108,109]. In parallel, interleukin-17 was reported to decrease nitric oxide bioavailability in smooth muscle cells and fibroblast, thereby promoting endothelial dysfunction and arterial stiffness [67]. The impact of dietary salt intake on the immune response has also been described in humans in a post-hoc analysis of a controlled simulated spaceflight program termed Mras520 [110]. Authors described an increase in inflammatory cytokines (IL-6 and IL-23) as well as a decrease in anti-inflammatory cytokines (IL-10) in the plasma of subjects submitted to high salt intake as compared to low salt intake. Recently, the gut microbiome has been proposed as a potential intermediary between sodium intake and clinical outcomes. The impact of a high sodium diet on gut microbial composition has been illustrated in various animal models [111,112]. In mice, chronic high sodium load induced qualitative and quantitative alterations in intestinal flora [113]. This study also revealed that sodium altered intestinal immunological gene expression and enhanced gut permeability as well as enteric bacterial translocation to the kidney. Excessive dietary sodium led to expansion of interleukin-17 producing T cells in the small intestine of mice, resulting in increased levels of circulating interleukin-17 [114]. High sodium intake affected mice gut microbiome by depleting Lactobacillus murinus in another study [115]. Consequently, treatment of those mice with L. murinus prevented salt-induced hypertension by modulating interleukin-17 producing T cells. Differences in gut microbial composition have been shown between salt-sensitive and salt-resistant Dahl rats and faecal transplantation from one strain to the other was able to change the BP pattern [116].

Altogether, the available evidence suggests that sodium not only affects target organs via its indirect effect on BP but, clearly, also through complex interconnected pathways involving oxidative, inflammatory, endocrine, immune and microbiological mechanisms.

## 4. Conclusions

The available evidence points toward a causal role of sodium intake on BP and cardiovascular prognosis. While the pathophysiological link between hypertension and cardiovascular events is relatively straightforward, a large body of data now suggests that sodium directly damages target organs independently of BP control via multiple intricate pathways. Although gaps in knowledge still exist, reduction in sodium intake on a population level represents a feasible strategy to reduce the burden of cardiovascular morbidity and mortality worldwide. 

## Figures and Tables

**Figure 1 nutrients-13-03177-f001:**
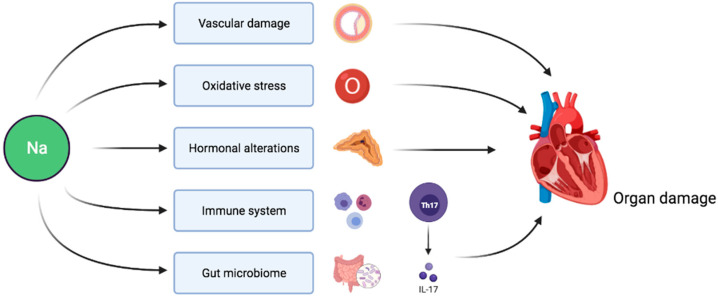
Pathophysiological pathways linking sodium intake to target organ damage independently of BP-mediated effects.

**Figure 2 nutrients-13-03177-f002:**
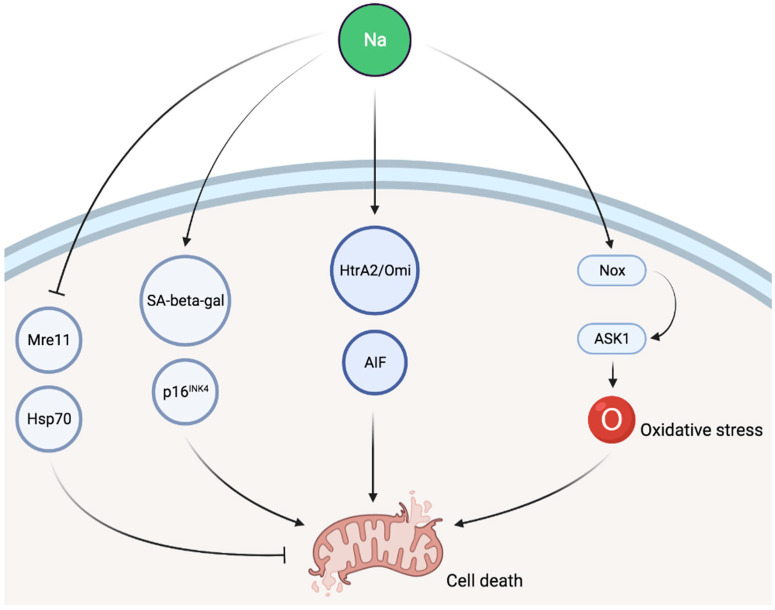
Schematic representation of potential pathways involved in sodium-induced cellular injury. SA-beta-gal, senescence-associated β-galactosidase; AIF, apoptosis-induced factor; Nox, NADPH oxidase; ASK1, apoptosis signal-regulating kinase 1.

**Figure 3 nutrients-13-03177-f003:**
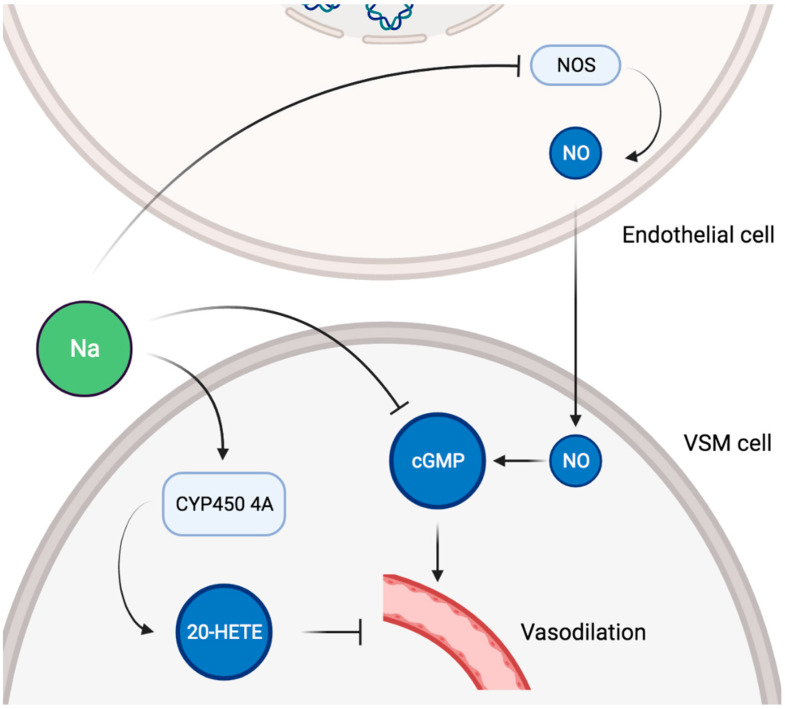
Schematic representation of potential pathways involved in sodium-induced vascular alterations. VSM, vascular smooth muscle; NOS, NO synthase; NO, nitric oxide; cGMP, cyclic GMP; 20-HETE, 20-hydroxyeicosatetraenoic acid.

**Table 1 nutrients-13-03177-t001:** Selected RCTs on sodium intake, blood pressure and cardiovascular outcomes.

Study	Population	Intervention	Outcome
**Sodium intake and blood pressure**			
DASH-sodium *n* = 412	Pre-HT; Age 47; Male 41%; White 40%	Sodium intakes of 3.3, 2.4 and 1.5 g/day	SBP reduction of 2.1 mmHg (3.3 vs 2.4 g/day) and 4.6 mmHg (2.4 vs 1.5 g/day) (*p* < 0.001)
TOHP-II *n* = 2382	Not HT; Age 43.9; Male 65.7%; White 79.3%	Sodium intake reduction to 80 mmol/day	DBP reduction of 0.7 mmHg (*p* = 0.10)
TONE *n* = 975	Treated HT; Age 65.8; Male 53%; White 76%	Sodium intake reduction to 80 mmol/day	SBP reduction of 3.4 mmHg (*p* < 0.001)
**Sodium intake and cardiovascular outcomes ^a^**			
Morgan et al. *n* = 77	Untreated HT; Age 57.1; Male 100%; White NA	Sodium intake reduction to 70–100 mmol/day	Relative risk of CV event: 1.16 (0.39–3.45)
TOHP-I *n* = 744	Not HT; Age 43.4; Male 71.4%; White 77.2%	Sodium intake reduction to 80 mmol/day	Relative risk of CV event: 0.51 (0.29–0.91)
TOHP-II *n* = 2382	Not HT; Age 43.9; Male 65.7%; White 79.3%	Sodium intake reduction to 80 mmol/day	Relative risk of CV event: 0.88 (0.65–1.20)
TONE *n* = 975	Treated HT; Age 65.8; Male 53%; White 76%	Sodium intake reduction to 80 mmol/day	Relative risk of CV event: 0.80 (0.53–1.21)

Abbreviations: RCT, randomized controlled trial; HT, hypertension; SBP systolic blood pressure: DBP, diastolic blood pressure; CV, cardiovascular. Age is expressed as mean. ^a^: Represented by follow-up studies including individuals who previously participated in RCT of sodium intake reduction. Data from Taylor et al. meta-analysis.

## Data Availability

Not applicable.

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
