# Peer review of "Sodium Intake as a Cardiovascular Risk Factor: A Narrative Review"

_nutrients, 2021, doi:10.3390/nu13093177_

Round 1

Reviewer 1 Report

The authors have presented a well written paper on Sodium as a cardiovascular risk factor. 

I must say I really enjoyed reading the paper. 

However,  a few additions would make this a well rounded article.

The authors need to review the molecular mechanisms of the pathophysiology of sodium. The last section seems rather hurried and not deep. For the benefit of the readers, it would be good to include some schema showing the molecular pathways that sodium effects within the cell.

Reviewer 2 Report

In this narrative review the authors describe, on basis of the most up-to-date knowledge, the relationship between sodium intake and health, with particular regard to hypertension and cardiovascular diseases, also focusing on the underlying pathophysiological mechanisms.

The manuscript is interesting and well written. The references are quite up to date. This reviewer raises only few issues that the authors have to address.

1- Intriguingly, both observational study (Diabetes Care 2006 Mar;29(3):498-503. doi: 10.2337/diacare.29.03.06.dc05-1776) and randomized clinical trial (Cardiovasc Diabetol 2021; 20:145. doi: 10.1186/s12933-021-01343-1) have suggested that sodium intake should be routinely evaluated in patients with diabetic nephropathy, a population at very high CV risk. This issue has a great clinical impact on management of these patients. Therefore, this point and the above referenced should be added by authors in this narrative review.

2- Figure 1 The mechanisms linking sodium intake to organ damage should be described more fully, in line with the contents of this review.

3- Table 1 should be better formatted, preferably horizontally, to facilitate reading.

4- The conclusions must be formatted like the text.
